# Identification of the Optimal Position of a Nasal Oxygen Cannula for Apneic Oxygenation: A Technical Simulation

**DOI:** 10.3390/jcm11226809

**Published:** 2022-11-17

**Authors:** Wolfgang A. Wetsch, Daniel C. Schroeder, Susanne J. Herff, Bernd W. Böttiger, Volker Wenzel, Holger Herff

**Affiliations:** 1Department of Anesthesiology and Intensive Care Medicine, University Hospital Cologne, Faculty of Medicine, University of Cologne, 50937 Cologne, Germany; 2Department of Anesthesiology and Intensive Care, German Armed Forces Central Hospital Koblenz, 56072 Koblenz, Germany; 3Department of Anesthesiology, Intensive Care Medicine, Emergency Medicine and Pain Therapy, Klinikum Friedrichshafen, 88048 Friedrichshafen, Germany; 4Department of Anesthesiology, University of Florida, Gainesville, FL 32611, USA; 5Department of Anesthesiology, PAN Clinic, 50667 Cologne, Germany

**Keywords:** airway manikin, apneic oxygenation, denitrogenization, desaturation, oxygen insufflation, test lung

## Abstract

Background: In a cannot-ventilate-cannot-intubate situation, careful preoxygenation with high FiO_2_ allowing subsequent apneic oxygenation can be life-saving. The best position for an oxygen supply line within the human airway at which oxygen insufflation is more effective than standard preoxygenation with a face mask is unknown. Methods: In this experimental study, we compared the effectiveness of preoxygenation by placing an oxygen cannula at the nose entrance, through the nose at the soft palatine, or at the base of the tongue; as a control we used ambient air. We connected a fully preoxygenated test lung on one side to an oximeter with a flow rate of 200 mL/min simulating the oxygen consumption of a normal adult on the other side of the trachea of an anatomically correctly shaped airway manikin over a 20 min observation period five times for each cannula placement in a random order. Results: The oxygen percentage in the test lung dropped from 100% in all groups to 53 ± 1% in the ambient air control group, to 87 ± 2% in the nasal cannula group, and to 96 ± 2% in the soft palatine group; it remained at 99 ± 1% in the base of the tongue group (*p* = 0.003 for the soft palatine vs. base of the tongue and *p* < 0.001 for all other groups). Conclusions: When simulating apneic oxygenation in a preoxygenated manikin, oxygen insufflation at the base of the tongue kept the oxygen percentage at baseline values of 99%, demonstrating a complete block of ambient air flowing into the airway of the manikin. Oxygen insufflation at the soft palatine or insufflation via a nasal cannula were less effective regarding this effect.

## 1. Background

During emergent or planned airway management, either the patient does not breathe sufficiently or anesthesia drugs have induced apnea. Thus, bag-valve-mask ventilation, followed by the insertion of an airway device and subsequent mechanical ventilation, has to be performed. If neither ventilation with bag-valve-mask ventilation, tracheal intubation, nor the placement of a supraglottic airway device (SGA) are successful, a cannot-ventilate-cannot-intubate situation (CVCI) occurs, with an imminent and severe risk of ensuing hypoxia and possibly death by suffocation [1].

Following preoxygenation for the scheduled induction of anesthesia, oxygen can be absorbed during apnea, thus significantly prolonging the time until hypoxia arises. However, moderate hypoxia often occurs during standard intubation attempts [2]. Recent studies have shown that the time until hypoxia occurs can be prolonged by diligent preoxygenation and employing apneic oxygenation, which may be life-saving. Different strategies to deliver oxygen have been described, such as specifically modified oxygenation laryngoscopes [3,4], oxygenation oropharyngeal tubes, or intratracheal oxygen insufflation with a bronchoscope [5,6,7]. In those studies, the deeper the oxygen was applied, the more effective it was, with an intratracheal application being the most effective [6,7]. However, we still do not know the optimum depth or the turning point in the human airway at which oxygen insufflation is more effective than nasal insufflation. Whilst intratracheal oxygen is most effective, it may be technically more difficult and may impair visibility and the ease of tracheal tube insertion during intubation attempts, especially in difficult airways whereas oxygen application at higher positions can easily be achieved by standard oxygenation devices, e.g., oxygenation laryngoscopes at the base of the tongue.

In order to assess the optimal position of oxygen insufflation in a human airway, we placed an oxygen cannula at several positions into the airway of a manikin that could easily be reached by standard oxygenation devices in an established technical model of apneic oxygenation and compared it with standard preoxygenation. The formal hypothesis was that there would be no difference in the oxygen percentage decrease between the groups.

## 2. Methods

Ethics approval was waived because this was solely a technical simulation with no participants or patients.

The trachea of a male intubation manikin (AirSim Combo Bronchi X, TruCorp, Lurgan, Northern Ireland) was connected to a test lung (volume: 3 L) representing the functional residual capacity of an adult man. An oximeter with a sampling rate of 200 mL/min (Wato Ex 35, Mindray, Shenzhen, China) was connected to the test lung. This represented the sampling rate equal to the oxygen uptake of a male adult during apneic oxygenation [8]. At the baseline, the airways and the test lung were filled with 100% oxygen. Insufflation of oxygen at a rate of 4 L/min was then performed using an oxygen cannula (Oxygen cannula, Asid Bonz, Herrenberg, Germany) placed: (1) at the nose entrance; (2) through the nose with the tip at the soft palatine; (3) at the base of the tongue (with the line placed through the oral cavity, comparable with an oxygenation laryngoscope). A control group received no oxygen insufflation. Five experiments for each group were performed in a random order, continuously measuring the decrease in oxygen percentage during a period of 20 min. The data were recorded every minute. The observer was blinded to the method of oxygen delivery.

The data were reported as the mean ± standard deviation. SPSS (IBM SPSS V28; IBM, Armonk, NY, USA) was used for the statistical analysis. After a Kolmogorov–Smirnov analysis, a post hoc analysis of variances was performed to determine the overall statistical significance between the groups. To assess the differences between the groups, a Student Newman–Keuls test was performed; *p* < 0.05 was considered to be statistically significant.

## 3. Results

During the 20 min observation period, the oxygen percentage in the test lung dropped from 100% at the baseline to 53 ± 1% in the control group, to 87 ± 2% in the nasal cannula group, and to 96 ± 2% in the soft palatine group. It remained at 99 ± 1% in the base of the tongue group (*p* = 0.003 for the soft palatine vs. base of the tongue and *p* < 0.001 for all other groups). The detailed course of the oxygen content decline in the test lungs for all groups is depicted in Figure 1.

## 4. Discussion

In this model of apneic oxygenation, oxygen insufflation at the base of the tongue (and thus, distal to the oral cavity) was the most effective in preventing environmental air to enter the airways and the lungs, thus maintaining apneic oxygenation.

Previous studies have also observed the superiority of deep oxygen insufflation and have attributed this effect to the anatomy of the kinked human airway: oxygen entering via a nasal cannula pours into the upper oral cavity and the upper pharynx, where it is likely to mix with nitrogen from environmental air entering through the mouth of the manikin [7]. The Venturi effect may emphasize this phenomenon by suctioning air into the airway, especially when higher oxygen flows are applied [9]. According to our data, this may, to a certain degree, also apply to oxygen given at the level of the soft palatine. In contrast, insufflating oxygen at the base of the tongue may be close enough to the trachea, thus completely preventing environmental air from entering the test lung.

In contrast to other studies favoring the application of very high oxygen rates for apneic oxygenation [10,11], we were able to show that apneic oxygenation could be maintained with a flow of only 4 L/min when applied distally, which was in accordance with a recently published study in humans [12]. An oxygen supply of 4 L/min exceeds the normal oxygen uptake of 200 mL/min being absorbed by the lungs (which was simulated by gas sampling in our model) 20-fold. This may build up a reservoir in the lower pharyngeal cavity, thus generating a flow of abundant oxygen back to the oral cavity that completely prevents ambient air from entering the airway.

A technical simulation is always a limitation by itself, especially regarding simulating apneic oxygenation. However, the most important points in this study were the anatomically correct shape of the human airway and the determination of the turning point from where no mixing with intruding nitrogen was evident; thus, we believed that this model, with an emphasis on correct anatomy, was sufficiently realistic to examine the hypotheses. Further, the principle of apneic oxygenation is well-known and gas consumption can be calculated [13,14,15]. We were not able to further simulate carbon dioxide production or the humidification of airway gases in this experiment. Thus, we were not completely realistic in simulating an apneic oxygenation scenario; e.g., carbon dioxide production may, after a period of time, interact with the oxygen transport in the blood on a cellular level and, over a long time, with the oxygen uptake in the lungs due to Dalton’s law that, in the end, limits apneic oxygenation. However, this may take hours, far longer than the time of this experiment [15]. Further, the most important precondition for apneic oxygenation is adequate preoxygenation in the test lung [16] and it is far easier to be achieved in this technical simulation compared with human lungs. In summary, our scenario may be realistic to demonstrate in an anatomically correct manner the point in the airway for oxygen insufflation that impairs ambient air flowing into the airway with weaknesses regarding the simulation of apneic oxygenation. More realistic scenarios regarding apneic oxygenation (such as animal experiments) would have meant testing with different anatomical airway shapes without a kinked airway, which was not possible for the purpose of this study. Studies in non-intubated humans may be unethical. Thus, in our opinion, this bench model was the best available method to test our hypothesis. Based on the actual study, the base of the tongue may be described as a turning point for oxygen insufflation in the human airway, completely avoiding any ambient air from entering the airway. This effect may be reached by different devices such as an oxygenation laryngoscope or a special oropharyngeal oxygenation airway device [3,4,17,18]. The value of the placement of these devices at the base of the tongue must be evaluated both in animal and clinical studies in the future.

## 5. Conclusions

When simulating apneic oxygenation in a preoxygenated manikin, oxygen insufflation at the base of the tongue kept the oxygen percentage at baseline values of 99%, demonstrating a complete block of ambient air flowing into the airway of the manikin. Oxygen insufflation at the soft palatine or insufflation via a nasal cannula were less effective regarding this effect.

## Figures and Tables

**Figure 1 jcm-11-06809-f001:**
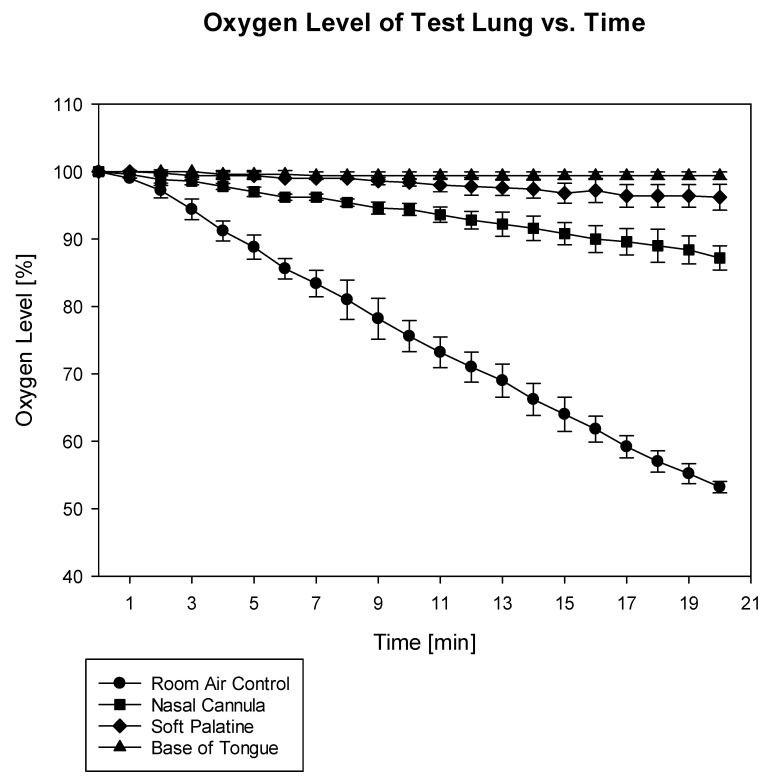
Course of oxygen content in the test lung for an observation period of 20 min (*p* = 0.003 for soft palatine vs. base of tongue and *p* < 0.001 for all other groups).

## Data Availability

All data are included in the manuscript. The original datasets analyzed during the current study are available from the corresponding author on reasonable request.

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
