# Peer review of "Identification of the Optimal Position of a Nasal Oxygen Cannula for Apneic Oxygenation: A Technical Simulation"

_jcm, 2022, doi:10.3390/jcm11226809_

Round 1

Reviewer 1 Report

I would like to congratulate the authors for the novelty of the idea and the quality of the data presented.  Despite the fact that the airway anatomy of experimental animals is different as mentioned by the authors I would like to see some data on this idea,  published in a following manuscript examining the effect of humidification and carbon dioxide exchange in relation to insufflation position. 

In the current manuscript the authors evaluated the use of different positioning of oxygen canula and the effect on lung oxygenation using an anatomically appropriate airway manikin. This is actually a question of high clinical importance for cases with difficult or impossible ventilation as by employing apneic oxygenation strategies we can importantly prolong the available time for airway manipulation permitting the use of advanced instrumentation.  Till recently the suggested approach included use of standard nasal cannulas (max 15 lt/min) which were partially replaced by high flow devices (60 lt/min) equipped with nasal cannulas. However, as authors mention these approaches may not be as effective as expected since atmospheric air is also “pumped” through venturi effect. The findings of this study may lead to guideline alterations as oxygen insufflation at the base of the tongue seems rather superior, increasing the available time to secure airway or to reverse neuromuscular blockade.

            One limitation of the study that could be elaborated more is that in the simulation manikin all gases are pumped out while in the real lung, only oxygen is absorbed. The findings would be greatly affected if the patient’s lung is not properly denitrogenated before the introduction of the canula. Furthermore, after prolonged apnea periods, the subsequent high carbon dioxide concentrations in the lung will physiologically decrease the available oxygen according to Dalton’s law, but probably at a non-clinically significant level.

Reviewer 2 Report

The authors compared the effectiveness of preoxygenation by placing an oxygen cannula at different anatomical positions within the airway in a male intubation manikin. They found that oxygen insufflation at the base of the tongue was the most effective. The experimental design is straightforward and logical. The methodology of this study is good. These findings are of interest, and this paper is well-written. However, the following comments should be addressed.

Minor comments:

1) In Figure 1 legends, it is unclear what symbol § representant. “§ p=0.003 for soft palatine vs base of tongue groups and <0.001 to all other groups.”

2) It states in the abstract background section that the best position for an oxygen supply line within the human airway at which oxygen insufflation is more effective than standard preoxygenation with a face mask, or comparably effective as intratracheal insufflation is unknown. However, the authors didn’t compare the effectiveness of preoxygenation by placing the oxygen cannula at several positions in the airway to intratracheal insufflation.

3) It seems that oxygen percentage during the 20-min observation period was recorded every minute instead of continuously. It’s better to mention this in the method.

4) The cannula tip positioned at the base of the tongue was through the nose or mouse? If through the nose, it would be interesting to add a group through the mouse for comparison. 
